# Improving Autoencoder Performance on Sparse Binary Data through Sparsity-Aware Loss Functions

## Abstract

Conventional reconstruction losses for autoencoders such as mean squared error (MSE) and binary cross-entropy (BCE) are poorly suited for sparse binary data. These measures can achieve deceptively low loss by trivially predicting the dominant zeros, while failing to capture the rare but informative non-zero entries. Prior work has primarily focused on architectural modifications or training heuristics to address this issue, leaving the design of loss functions largely overlooked. In this work, we shift focus to the reconstruction loss itself, exploring sparsity-aware reconstruction losses by extending focal loss, dice loss, and related formulations to the autoencoder setting. We evaluate their effect on both reconstruction fidelity and embedding quality across multiple sparse datasets, showing that these alternatives outperform MSE and BCE on metrics sensitive to rare events. Our results demonstrate that the choice of loss function is a critical but underappreciated factor in learning effective representations from sparse binary data.

## 1 Introduction

Autoencoders (AEs) are a standard tool for representation learning, dimensionality reduction, and generative modeling. Their effectiveness, however, depends critically on the reconstruction loss, which determines what aspects of the data the model prioritizes. Standard objectives such as mean squared error (MSE) and binary cross-entropy (BCE) are poorly aligned with the challenges of sparse binary data: because zeros dominate, a model can minimize loss by trivially predicting the majority class, suppressing the rare but informative positives that drive downstream performance.

While previous work has acknowledged the difficulty of training autoencoders on sparse data, most approaches have targeted architectural or procedural modifications—such as denoising (Vincent et al. (2008)), masked autoencoders (He et al. (2022)), and variational formulations (He et al. (2022)). In contrast, the design of reconstruction losses has received little attention.

We address this gap by adapting imbalance-aware objectives that have proven effective in domains such as object detection and medical image segmentation. Specifically, we extend focal loss (Lin et al. (2017)), Dice loss (Sudre et al. (2017)), and weighted variants of MSE/BCE to the autoencoder setting, and systematically evaluate their impact on both reconstruction fidelity and representation quality.

Our contributions are as follows:

1. A formal analysis of how sparsity affects standard reconstruction objectives, highlighting their insensitivity to rare positives.

2. Adaptation of sparsity-aware loss functions to the autoencoder setting as a simple, general alternative to architectural modifications.

3. An empirical study across multiple sparse datasets, assessing reconstruction fidelity, embedding structure, and downstream task performance.

## 2 RELATED WORK

Autoencoders typically minimize MSE or BCE; the latter can be interpreted as a Bernoulli negative log-likelihood (Kingma & Welling (2013)). To handle sparsity, prior work has emphasized architectural or procedural variants, including denoising autoencoders (Vincent et al. (2008)) and masked autoencoders (He et al. (2022)).

In contrast, other machine learning domains have emphasized loss design as a key mechanism for handling imbalance. The focal loss (Lin et al. (2017)), introduced for dense object detection, down-weights abundant "easy negatives" and increases the contribution of rare positives, improving learning under extreme imbalance. Dice loss (Sudre et al. (2017)) directly optimizes overlap between prediction and ground truth. These examples demonstrate how tailoring loss functions to reflect data imbalance can substantially improve performance.

Reconstruction losses also shape the geometry of the learned latent space. It is well-established that objectives defined in a feature space, rather than pixel space, produce representations that better align with semantic and perceptual similarity. Johnson et al. (2016) first showed that replacing pixel-wise objectives with "perceptual" losses derived from a pretrained network yields superior results in image transformation tasks. More recent self-supervised methods have taken this concept further by discarding pixel reconstruction entirely; contrastive frameworks (Chen et al. (2020)) learn powerful representations by optimizing an objective solely within the embedding space. This dual role of the objective, governing both reconstruction and representation, suggests that imbalance-aware losses may benefit autoencoders beyond per-entry fidelity.

Despite these advances in other fields, autoencoder research on sparse binary data has largely overlooked loss design. Our work bridges this gap by systematically evaluating focal loss, Dice loss, and weighted reconstruction objectives in this context.

## 3 LOSS AND SPARSITY

Sparse binary datasets pose unique challenges for autoencoder reconstruction. To analyze this, we first formalize sparsity, its impact on standard loss objectives, and then introduce sparsity-aware alternatives.

### 3.1 MEASURING SPARSITY

Let $X \subset \{0,1\}^{n \times d}$ denote a binary dataset with $n$ samples and $d$ features. For a feature $j \in [d]$, the probability of observing a 1 is $p_j$ and the probability of observing a 0 $s_j$ as defined by:

$$p_j = \frac{1}{n} \sum_{i=1}^{n} X_{ij},$$

$$s_j = 1 - p_j \tag{1}$$

The overall dataset sparsity is the average across features:

$$s(X) = \frac{1}{d} \sum_{j=1}^{d} s_j \tag{2}$$

### 3.2 STANDARD RECONSTRUCTION LOSSES

For an autoencoder reconstruction $\widehat{x} \in \widehat{X}$ from input $x \in X$, the expected loss for MSE and BCE is:

$$E[\text{MSE}] = \frac{1}{d} \sum_{j=1}^{d} \left( p_j (1 - \widehat{x}_j)^2 + s_j \widehat{x_j^2} \right) \tag{3}$$

$$E[\text{BCE}] = -\frac{1}{d} \sum_{j=1}^{d} \left( p_j \log \widehat{x}_j + s_j \log (1 - \widehat{x}_j) \right) \tag{4}$$

As sparsity increases ($s_j \to 1$, $p_j \to 0$), $\widehat{x}^2$ term dominates for MSE and $\log(1 - \widehat{x}_j)$ term dominates for BCE. Thus, both losses become insensitive to rare positives, by rewarding trivial all-zero predictions.

### 3.3 Sparsity-Aware Losses

A natural first step to counter imbalance is to upweight the positive terms directly, yielding weighted variants:

- WeightedMSE (WMSE):

$$\text{WMSE}(x, \hat{x}) = \frac{1}{d} \sum_{j=1}^{d} [\alpha(1 - \widehat{x}_j)^2 + (1 - x_j)\widehat{x}_j{}^2] \tag{5}$$

- WeightedBCE (WBCE):

$$\text{WBCE}(x, \hat{x}) = -\frac{1}{d} \sum_{j=1}^{d} [\alpha(x_j \log \widehat{x}_j) + (1 - x_j) \log(1 - \widehat{x}_j)] \tag{6}$$

Here, $\alpha$ scales positive contributions. We define $\alpha = \beta + \log \frac{1}{p_j}, \beta > 1$.

Focal loss (Lin et al. (2017)) further addresses imbalance, adapting BCE to dynamically scale the loss of easy-to-predict examples:

$$\text{FocalLoss}(x, \hat{x}) = -\frac{1}{d} \sum_{j=1}^{d} \left[ \alpha(1 - \widehat{x}_j)^\gamma x_j \log \widehat{x}_j + (1 - \alpha) \widehat{x_j^\gamma} (1 - x_j) \log(1 - \widehat{x}_j) \right] \tag{7}$$

Where $\gamma > 0$ is the focusing parameter. The focusing parameter reduces the weight of confident predictions, so that rare positives and harder cases dominate the gradient. For example, if $x = 0$ reconstructed correctly with high confidence ($\hat{x} = 0.01$), its contribution under BCE is $log(0.99) \approx$ –0.004, but with focal loss and $\gamma = 2$ the term is scaled by $(0.01)^2 = 0.0001$, making it 100× smaller. This shift suppresses the overwhelming influence of easy zeros, allowing rare positives to guide learning.

Finally, Dice loss (Sudre et al. (2017)) originates in segmentation tasks with extreme class imbalance:

$$\text{DiceLoss}(x, \hat{x}) = 1 - \frac{2 \sum_{j=1}^{d} x_j \widehat{x}_j + \epsilon}{\sum_{j=1}^{d} x_j + \sum_{j=1}^{d} \widehat{x}_j + \epsilon} \tag{8}$$

Dice directly optimizes overlap between predictions and true positives, ignoring true negatives entirely. While this ensures rare positives remain influential, gradients may vanish at extreme sparsity levels.

## 4 Experimental Setup and Results

To assess our claims experimentally, we train AEs for each dataset on each loss. For all experiments, we keep the architecture and hyperparameters of the AE fixed, as detailed in Appendix B, varying only the reconstruction loss.

### 4.1 Datasets

We select 5 binarized datasets that vary in sparsity ($s \approx 0.85$–$0.999$), enabling us to test whether sparsity-aware losses provide consistent benefits across both extreme and moderate sparsity regimes. Table 1 (Appendix A) reports full statistics.

- Netflix (three variants) ($s \approx 0.97, 0.98, 0.999$): Derived from the Netflix Prize dataset, binarized to indicate whether a user has consumed an item (Netflix Inc. (2016)).

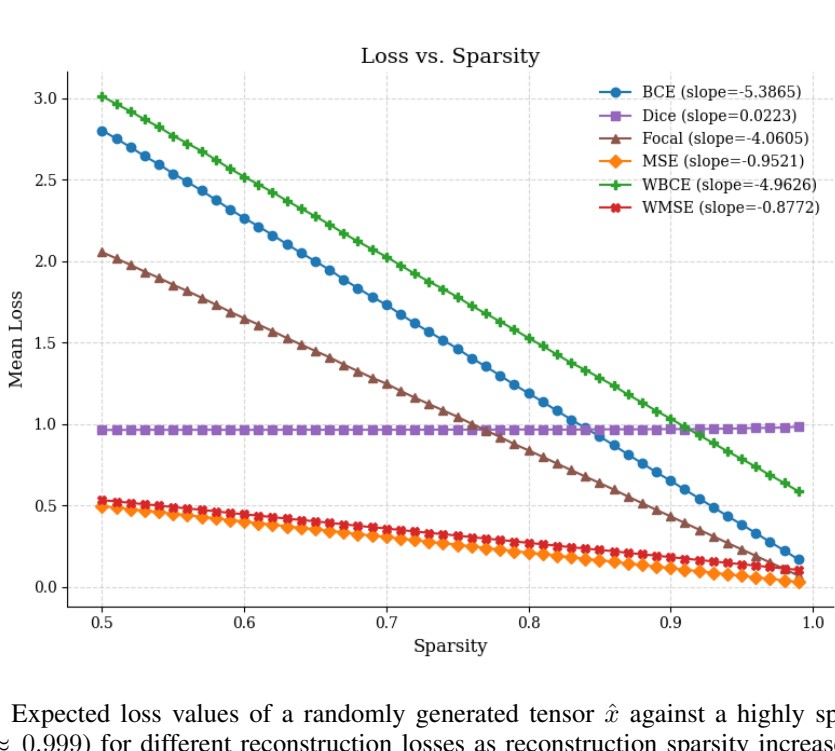

Figure 1: Expected loss values of a randomly generated tensor $\hat{x}$ against a highly sparse target $x$ ($s(x) \approx 0.999$) for different reconstruction losses as reconstruction sparsity increases ($s(\hat{x}) \in [0.5, 0.999]$). MSE/BCE degrade faster with sparsity, as compared to their modified variants. Dice loss remains largely invariant to sparsity.

- IMDB ($s \approx 0.999$): One-hot encoded content metadata (language, country, content type). Downstream task: predict real-valued IMDB ratings (Narayan (2022)).

- Rheumatic ($s \approx 0.85$): Binary symptom questionnaires from an online symptom checker. Downstream task: predict self-reported outcomes (SpA, PMR, APs) ("Rheumatic?" (2024)).

## 4.2 EVALUATION METRICS

To assess the claim that sparsity-aware losses can improve autoencoder performance, we evaluate across 3 criteria:

1. Reconstruction quality:

    (a) Average Precision (AP): Evaluating reconstruction against inputs computed on reconstructions compared to the original data.

    (b) Non-trivial Pairwise Contingency Score: Evaluating whether non-trivial feature dependencies are preserved by comparing normalized contingency tables between the original and reconstructed data[1]. The adapted score (Dat (2025)) is restricted to rows with at least one non-zero entry. For real data $R$ and reconstructed data $S$, the score is defined as follows:

$$\mathcal{I} = \{i \mid R_{i,A} \neq 0 \text{ or } R_{i,B} \neq 0\}$$
$$\text{ContingencyScore} = 1 - \frac{1}{2} \sum_{\alpha \in A} \sum_{\beta \in B} \left| S'_{\alpha,\beta} - R'_{\alpha,\beta} \right| \quad (9)$$

2. Embedding Structure:

    (a) Calinski–Harabasz (CH) Score (Caliński & Harabasz (1974)): Measures clustering quality applied in the embedding space, by balancing intra-cluster compactness and inter-cluster separation.

---

[1]For the contingency score, the outputs of the models were binarized using a threshold $t = 0.5$

     i. CH Score (DBScan clusters): Computed using cluster labels assigned by DBScan (Ester et al. (1996)), a density-based clustering algorithm Higher scores indicate that embeddings yield clusters that are both compact and well separated.

     ii. CH Score (sparsity groups): Same score, but applied to clusters defined by per-sample sparsity (fraction of zeros per input). This evaluates whether embeddings preserve the natural grouping induced by sparsity levels.

  (b) Trustworthiness (Venna & Kaski (2001)): Quantifies local neighborhood preservation between input and embedding spaces, measured by comparing neighborhoods in the original space versus the learned embedding. A high score indicates that nearest neighbors in the original space remain close in the embedding, reflecting faithful retention of fine-grained relationships.

3. Downstream Utility

  (a) Regression: Linear regression on embeddings for predicting ratings on the IMDB dataset, evaluated with $R^2$.

  (b) Classification: Logistic regression on embeddings for predicting self-reported diagnosis on the Rheumatic dataset, evaluated with Average Precision (AP) across diagnostic labels.

## 4.3 RESULTS

### 4.3.1 RECONSTRUCTION QUALITY

Sparsity-aware losses better preserve feature dependencies, while MSE and BCE achieve higher AP (Fig. 2). The reconstruction distributions of MSE and BCE are more sharply concentrated near 0 (Fig. 3), leading to fewer ambiguous mid-range values and thus more decisive rankings when computing AP. By contrast, sparsity-aware losses place more probability mass in the intermediate region, specifically in the region where $\hat{x} > t = 0.5$, which lowers the AP but improves the Pairwise Contingency Score (Fig. 2b), indicating better preservation of dependencies between features. Notably, all models achieve their highest contingency scores on the IMDB dataset, likely because its one-hot metadata features form strong, low-noise dependencies (e.g., language–country pairs) that are easier to reconstruct even under high sparsity.

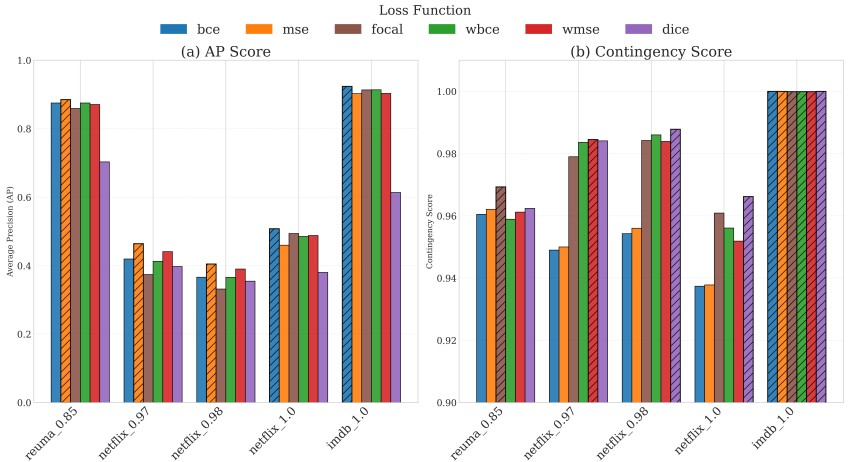

Figure 2: Reconstruction quality across datasets and sparsity levels. The best performing losses are marked with stripes for each dataset and metric. (a) Average Precision scores and (b) Pairwise Contingency scores for AE reconstruction using different loss functions.

### 4.3.2 EMBEDDING STRUCTURE

Sparsity-aware losses have better results on the embedding space metrics when compared to MSE or BCE (Fig. 4). Focal loss is particularly effective, balancing local neighborhood preservation (trust-

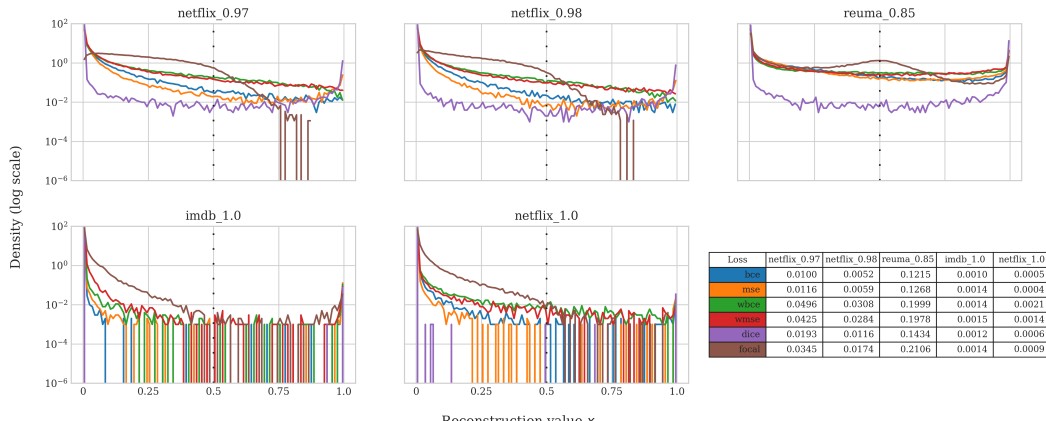

Figure 3: Distribution of reconstruction values across different loss functions on multiple datasets. Each curve shows the histogram density (log-scaled) of per-entry reconstructions, with the dashed line marking the threshold $t = 0.5$. The legend reports the mass right to the threshold. As sparsity increases, the distribution across experiments becomes less smooth. In high sparsity datasets ($s \approx 1$), Dice only predicts values close to 0 or 1.

worthiness) with global cluster separation (CH score). WBCE and WMSE also perform strongly and reliably across both metrics, while Dice emphasizes global grouping but can be less stable for local structure. The IMDB dataset naturally clusters by metadata categories (e.g., language–country pairs), which align closely with sparsity patterns. This makes sparsity-defined clusters more separable in the embedding space, boosting CH scores across all losses.

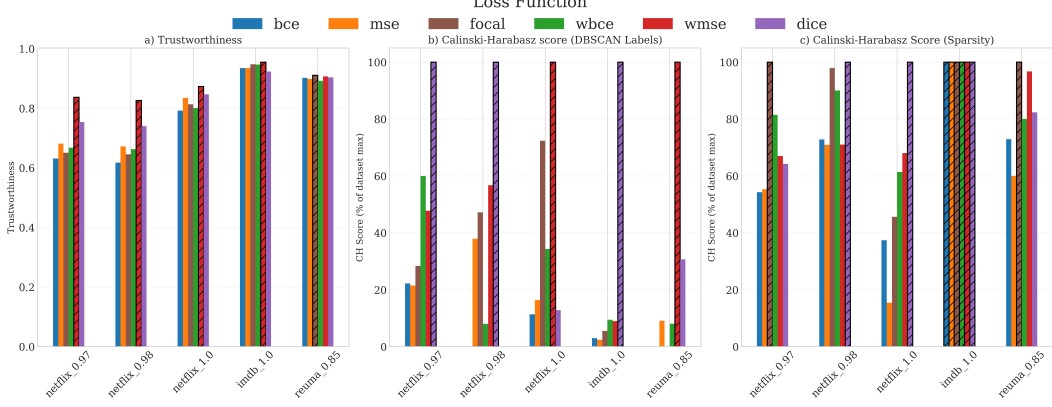

Figure 4: Embedding space evaluation across losses. The best performing losses are marked with stripes for each dataset and metric. a) Trustworthiness: Scale from 0 (worst) to 1 (best). b) CH Score on labels from unsupervised clustering (DBScan) of the embedding space, score is scaled from 0 to 100, each value is represented as a percentage of the maximum score achieved for its respective dataset. c) CH Score on sparsity labels, where the labels are generated by calculating how sparse a sample is. Scaled similarly to b).

### 4.3.3 DOWNSTREAM UTILITY

Sparsity-aware losses, specifically WBCE and WMSE, outperform BCE and MSE across both regression ($R^2$) and classification (AP) tasks (Fig. 5). They yield embeddings that generalize better to real-valued prediction and capture label-specific structure more effectively.

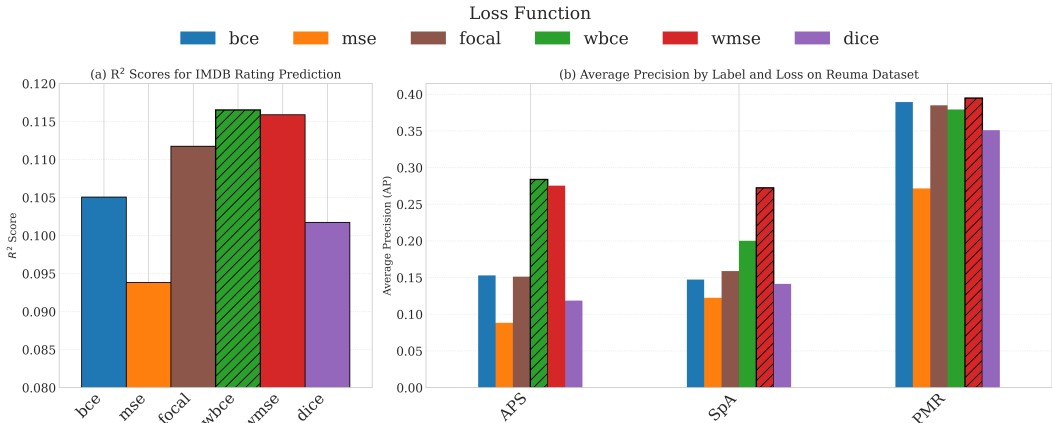

Figure 5: Downstream performance on (a) IMDB, fitting a linear regression model on embeddings to predict ratings. (b) Reuma fitting a logistic classifier on embeddings to predict self reported diagnosis of APS (Antiphospholipid Syndrome), SpA (Spondyloarthritis) and PMR (Polymyalgia Rheumatica). The best performing losses are marked for each dataset.

| | Reconstruction Quality | | Embedding Structure | | | Downstream Tasks | | | |
|---|---|---|---|---|---|---|---|---|---|
| | AP | Pairwise Contingency | Trustworthiness | CH Score (DBScan) | CH Score (Sparsity) | R2 Score Rating (Regression) | AP SpA (Prediction) | AP PMR (Prediction) | AP Aps (Prediction) |
| Netflix 0.97 | MSE | WMSE | WMSE | DICE | Focal | | | | |
| Netflix 0.98 | MSE | Focal | WMSE | DICE | Dice | | | | |
| Netflix 1.0 | BCE | DICE | WMSE | WMSE | Dice | | | | |
| IMDB 1.0 | BCE | All | WMSE | Dice | All | WBCE | | | |
| Reuma 0.85 | MSE | Focal | Focal | WMSE | Focal | | WBCE | WMSE | WMSE |

Figure 6: Summary of which reconstruction loss achieves the best performance for each dataset–metric combination. Each cell marks the top-performing loss, allowing direct comparison across reconstruction, embedding, and downstream tasks.

## 5 CONCLUSION

This work demonstrates that, for AEs trained on sparse binary data, the reconstruction loss itself provides a critical inductive bias. The key insight is not the discovery of a single loss function that universally outperforms MSE or BCE, but rather that incorporating sparsity-awareness into the objective consistently improves both reconstruction and representation quality.

Empirically, we find that MSE and BCE achieve the strongest per-entry recovery (AP), but systematically underperform on relational metrics, embedding structure, and downstream utility. In contrast, sparsity-aware losses (WBCE, WMSE, Focal, Dice) yield embeddings that are more trustworthy, better clustered, and more effective for prediction tasks (Fig. 6). These results suggest that loss functions emphasizing rare positives implicitly encourage the model to preserve higher-order dependencies that are overlooked by elementwise objectives.

The relatively modest margins observed are not surprising: all objectives optimize reconstruction, and with sufficient capacity, AEs can fit sparse data regardless of the loss. However, the value of sparsity-aware objectives lies in their consistent inductive bias, observed across datasets ranging from extreme sparsity ($s \approx 0.999$) to moderate sparsity ($s \approx 0.85$). By amplifying rare but in-

formative signals, they steer representations toward capturing both local dependencies and global structure.

From a theoretical perspective, these findings confirm that loss functions can shape the geometry of the embedding space as much as they affect reconstruction fidelity (Johnson et al. (2016)). In sparse regimes, objectives that overweight positives act as regularizers against trivial all-zero solutions, redistributing gradient signal across rare but meaningful events.

Looking forward, we expect larger performance gaps to emerge when sparsity-aware objectives are paired with complementary advances. In future work, we will extend sparsity-aware losses to sparse non-binary domains (e.g., sparse count matrices).

## REPRODUCIBILITY

All datasets used in this work are cited with details listed in Appendix A. The details of the autoencoders and experiments are listed in Appendix B. The details of the evaluation experiments are listed in Appendix C.

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

## A    APPENDIX: DATASETS

Table 1: Benchmark dataset descriptions.

| Dataset | Sparsity | # of Features | Train Size | Test Size | Val. Size | Downstream Task |
|---|---|---|---|---|---|---|
| Netflix_1.0 | 0.9987 | 4498 | 144827 | 48277 | 48278 | N/A |
| Netflix_0.97 | 0.97 | 470 | 34600 | 4325 | 4324 | N/A |
| Netflix_0.98 | 0.98 | 675 | 25840 | 3230 | 3231 | N/A |
| IMDB_1.0 | 0.99965 | 2265 | 7164 | 895 | 896 | Regression: Predicting ratings on IMDB |
| Reuma_0.85 | 0.85 | 488 | 16136 | 2017 | 2017 | Prediction: Predicting self-reported outcomes |

Table 2: Descriptions for benchmark datasets. See Table 1 for additional information.

| Dataset | Description |
|---|---|
| Netflix_1.0 | Binarized version of the Netflix Prize dataset. Rows are users, columns are content, a 1 indicates if a user has watched the content. |
| Netflix_0.97 | Filtered version of Netflix_1.0. Features and samples are iteratively removed at random to reduce sparsity. |
| Netflix_0.98 | Similar to Netflix_0.97. |
| IMDB_1.0 | Binarized dataset of content descriptions. Features are one hot encoded for language, country and content type. |
| Reuma_0.85 | Questionnaire responses from a symptom checker. Rows are users, columns are yes or no questions. |

## B    APPENDIX: EXPERIMENT DETAILS

Parameters for focal loss: $\alpha = 0.75$, $\gamma = 2$

Hyper-Parameters for autoencoders (shared across all experiments):

- **Input/Output layers**: Match the dimensionality of the dataset.
- **Activation functions**: ReLU for all layers except the final output layer, which uses a sigmoid activation.
- **Architecture**:
  - Encoder hidden layer: 200 units
  - Latent dimension: 50
  - Decoder hidden layer: 200 units
- **Normalization**: Batch normalization applied between layers.
- **Optimizer**: Adam with a learning rate of $1 \times 10^{-6}$.
- **Initialization**: Xavier uniform initialization.

All models were trained on the train split of the datasets, with early stopping based on the validation split. All evaluations were done on the test split of the respective datasets.

# C APPENDIX: EVALUATION DETAILS

This appendix provides additional details on how the evaluation metrics described in Section 4.2 were applied in our experiments.

## C.1 RECONSTRUCTION QUALITY

- **Average Precision (AP):** Reconstructions were treated as probabilistic predictions of binary inputs. Each input dimension was evaluated as a binary classification problem, with AP computed by ranking reconstructed values against the ground truth.

- **Pairwise Contingency Score:** Reconstructions were binarized with threshold $t = 0.5$. For each dataset, we computed normalized contingency tables between original and reconstructed features, restricted to rows with at least one non-zero entry. The final score is 1 minus the $\ell_1$ distance between normalized contingency tables.

## C.2 EMBEDDING STRUCTURE

- **Trustworthiness:** Computed by comparing $k$-nearest neighbors ($k = 15$) in the original input space versus the learned embedding space. Higher scores indicate greater preservation of local neighborhoods.

- **Calinski–Harabasz (CH) Score:**
  - *DBScan clusters:* CH scores were computed on cluster labels obtained by DBScan applied to the embedding space.
  - *Sparsity groups:* CH scores were computed on clusters defined by per-sample sparsity. For each dataset, the sparsity of each sample was calculated as the fraction of zero entries, and samples were then binned into 10 equally spaced groups.

## C.3 DOWNSTREAM UTILITY

- **Regression (IMDB):** A linear regression model was trained on embeddings to predict real-valued IMDB ratings. Performance was measured by $R^2$.

- **Classification (Rheumatic):** Logistic regression was trained on embeddings to predict diagnostic outcomes (APS, SpA, PMR). Performance was measured by Average Precision (AP) across labels.

