# OpenReview forum: "Improving  Autoencoder Performance on Sparse Binary Data through Sparsity-Aware Loss Functions"
_ICLR.cc/2026/Conference — ICLR 2026 Conference Withdrawn Submission_

### Official Review · Reviewer_JLiz · 2025-10-26

**Soundness:** 2
**Presentation:** 3
**Contribution:** 1
**Rating:** 2
**Confidence:** 4

**Summary:**

The paper presents the results of an extensive experimental study in which the authors seek to understand which existing losses for training autoencoders are more suitable (according to various evaluation criteria) in a setting where the inputs are binary and sparse.

**Strengths:**

- The experimental section considers multiple different AE methods and a broad range of evaluation metrics.
- Some of the experimental results could be important to a relevant audience, provided that they are presented and motivated appropriately.

**Weaknesses:**

- The motivation for the setting considered in the paper could be improved. In particular, it is unclear why obtaining a sparse latent representation is important. The only two downstream tasks being considered (linear regression and classification) can be adapted to work well with sparse data by incorporating sparsity-inducing regularization.
- The paper should be better positioned in the existing literature. Currently, the manuscript has a rather limited literature review of about 15 references, despite the area of (sparse) representation learning being a rich and active area of research.
- The extensive experimental results could be presented in an easier-to-digest way that highlights more clearly what the main takeaways are. Currently, the main contribution seems to be the conclusion that some losses are better suited for certain evaluation criteria, which is a somewhat expected outcome of any extensive empirical study such as the one presented in this paper. It would help if the paper formulated hypotheses about what the likely causes for the mixed results presented in Table 6 are.

**Minor issues**

- Some of the claimed contributions are not sufficiently justified. For example, contribution 1 (lines 47-48) claims “a formal analysis of how sparsity affects standard reconstruction objective”, however, the paper only presents empirical analyses (no formal statements). Moreover, in line 151 it is stated that “To assess our claims experimentally, we train AEs for each dataset on each loss”. However, it is unclear at this point what the claims are.
- Figure 1 is never referenced in the text. Moreover, its setting is never thoroughly described.

**Questions:**

- How do the numbers in figure 5 compare to running LASSO linear/logistic regression in input space?
- Are there any consistent phenomena revealed by the rather extensive experiments that were run for the paper? For instance, are there classes of losses that consistently show certain behaviors, and what are the explanations for them? Can additional ablations or theoretical analyses suggest potential causes for some of the observations in Table 6?

---

### Official Review · Reviewer_6a69 · 2025-10-30

**Soundness:** 2
**Presentation:** 2
**Contribution:** 1
**Rating:** 2
**Confidence:** 4

**Summary:**

The current paper studies the application of Dice loss (Sudre et al. (2017)) and Focal loss (Lin et al. (2017)) together with weighted variants of Binary Cross-Entropy (BCE) and Mean Squared Error (MSE) within the context of training autoencoders on sparse binary data. Evaluations are performed on the Netflix, IMDB and Rheumatic datasets to assess the impact of each loss in terms of the quality of the reconstructions, the embedding structure and the impact upon downstream performance.

**Strengths:**

- The paper is clearly written and easy to follow
- The evaluation features multiple criteria and multiple metrics are employed for each criteria. Average Precision and the Non-trivial Pairwise Contingency Score are used to measure reconstruction quality. The embedding structure is evaluated in terms of trustworthiness and the Calinski–Harabasz Score. Downstream utility is measured for both classification and regression tasks.

**Weaknesses:**

The main weakness of the current work is the lack of novelty. As it stands, the paper presents a straight-forward evaluation of existing losses specifically designed for the task at hand, i.e. to address significant imbalances. Given the fact that the paper does not propose a novel approach (in terms of either the loss function, the architecture or the training regime) a new challenging and realistic benchmark that could be interesting or useful for the community and because it does not present any new insights (i.e. the chosen losses are expected to work well in these scenarios are they are specifically designed to address imbalances), the work is not competitive with the existing ICLR submissions.

**Questions:**

- Can the authors propose a new loss function?
- Can the authors describe the revelance of sparse binary autoencoders within the current literature?
- Are there any new benchmarks?

---

### Official Review · Reviewer_do7a · 2025-10-31

**Soundness:** 1
**Presentation:** 1
**Contribution:** 1
**Rating:** 0
**Confidence:** 3

**Summary:**

The paper performs a comparison of different sparsity-aware loss function for autoencoders using binary input features.

**Strengths:**

N/A

**Weaknesses:**

1. I don't understand the contribution of the paper. In its current form, the paper simply performs a comparison of different loss functions that are well-known in the literature.

2. The motivation behind the problem setup is unclear. Why are binary input features an important setting for autoencoders? The experimental setting doesn't provide any insight into this issue as only classification datasets are explored. The paper doesn't provide comparison with classifiers.

3. Why is this task important to study in the current regime of powerful models? Each of these tasks could be easily solved using LLMs or embedding-based retrievers. The paper should provide comparison with the state-of-the-art methods in each setting.

**Questions:**

1. What is the proposed method in this paper? Currently, it appears to be an ablation study of different losses.
2. The paper should provide clarification about what tasks necessitate the use of autoencoders using binary features. What are the limitations of existing machine learning frameworks in solving such tasks?

---

### Official Review · Reviewer_DyiF · 2025-11-01

**Soundness:** 2
**Presentation:** 2
**Contribution:** 1
**Rating:** 2
**Confidence:** 3

**Summary:**

This paper aims to modify reconstruction losses for sparse data reconstruction. The authors analytically show that BCE and MSE are not well-suited for highly sparse data due to the imbalance between class 0 and 1. They introduce class weighting for MSE and BCE, and also explore the use of Focal Loss and Dice Loss for this problem. Experiments are conducted using autoencoders to reconstruct five different high-sparsity datasets (derived from three datasets but with different sparsity thresholds). Results show a trade-off between sparsity-aware losses and MSE/BCE in terms of average precision score and contingency score. Experiments also indicate that sparsity-aware losses are better for downstream tasks that use embeddings from the autoencoder models.

Claimed Contributions:

1. Demonstrates that BCE/MSE are not suitable for sparse datasets.

2. Introduces class weighting and the use of Focal & Dice Loss for sparse data reconstruction.

3. Experiments on sparse datasets.

**Strengths:**

1. The paper addresses a well-known and meaningful problem.

2. Figure 6 clearly highlights which losses are best for which tasks, improving readability, and can be useful for other researchers for their own research!

**Weaknesses:**

0. This is a well-known problem, and many existing works already address class imbalance using weighting schemes, L1 norms, etc. The paper lacks comparison to these existing methods; related work and experimental comparisons are very limited.

1. Limited novelty: using class weighting and well-known losses like Focal and Dice Loss are minor modifications.

2. Unclear goal: The paper title suggests "improving autoencoder performance," but experiments show that sparsity-aware losses have worse average precision. Downstream utility is better for Weighted MSE, but it is arguable whether this truly improves autoencoder performance. The authors should demonstrate strong performance across multiple aspects of autoencoders.

3. Unclear contribution: Weighting schemes and Focal/Dice losses are already well-known solutions.

4. Limited dataset variety: Only three datasets are tested, and results are not very strong.

**Questions:**

I have one main question for the authors: It is unclear how the embeddings from the autoencoder models are used for downstream tasks. Are they taken from the latent space?

---

### Note · Authors · 2025-11-26

I have read and agree with the venue's withdrawal policy on behalf of myself and my co-authors.